# PiLoRA: Gradient-Informed Parameter-Importance-Aware Low-Rank Adaptation

## Abstract

Parameter-Efficient Fine-Tuning methods, such as Low-Rank Adaptation (LoRA), are widely used to adapt pre-trained models (PTMs) due to its computational efficiency. However, LoRA and existing LoRA-based methods treat weight modules in the PTMs as indivisible units and apply a uniform adaptation capacity. This coarse, module-level approach overlooks the varying importance of parameters within a module and fails to exploit the inherent structured sparsity of foundation models. To address these limitations, we propose **P**arameter-**I**mportance-Aware **Lo**w-**R**ank **A**daptation (PiLoRA), a novel PEFT method that allocates different adaptation capacity based on parameter importance. Our approach uses efficient low-rank gradients to approximate parameter importance in the full weight parameter space, enabling partition of neurons into distinct groups. Important neurons receive a higher adaptation capacity for targeted training, while others are tuned with lower capacity for efficiency. Comprehensive experiments on both the language reasoning and image classification settings demonstrate that PiLoRA consistently outperforms vanilla LoRA with targeted fine-tuning using fewer parameters, striking a balance between effective adaptation and efficiency. Our work shifts the focus of adaptation from the module level to a more granular neuron level and unlocks a more powerful and efficient approach to PEFT.

## 1 Introduction

Large Language Models (LLMs) have achieved great success across a wide range of domains, from natural language understanding and generation to complex reasoning tasks (Touvron et al., 2023; Achiam et al., 2023; Dai et al., 2024). However, adapting these large-scale models to downstream tasks remains a challenging problem, as full fine-tuning requires retraining billions of parameters, incurring significant computational costs (Dodge et al., 2020). Parameter-efficient fine-tuning (PEFT) methods have emerged as a promising solution, enabling downstream adaptation by updating only a small subset of parameters (Houlsby et al., 2019; Li & Liang, 2021). Among these, LoRA (Hu et al., 2021) has become one of the most widely adopted techniques, due to its simplicity and effectiveness. LoRA injects low-rank, trainable matrices into the model while keeping the pre-trained weights frozen, which substantially reduces training costs while preserving adaptation performance.

Despite its success, LoRA and its variants typically treat all parameters (e.g., the weight matrix of a linear layer) within a module (e.g., the attention or feed-forward layers) as a whole, updating them jointly via assigning the same rank regardless of their relative importance. This uniform approach often leads to suboptimal results for two key reasons: (1) It overlooks the varying importance of different parts of the weight matrices, causing the ineffective usage of the low-rank adaptation capacity. While some methods (Zhang et al., 2023; Ding et al., 2023; Chang et al., 2025) adaptively optimize the ranks of each LoRA component and allocate the parameter budget for different modules, they are restricted treat each component as an indivisible unit. This neglects the fine-grained importance of the parameters within each module, leading to inefficient adaptation capacity allocation. (2) It fails to leverage the inherent structured sparsity present in foundation models. It is well-established that not all neurons or parameter groups contribute equally to a model's capabilities and structured sparsity can be used to improve the training and inference efficiency (Liu et al., 2023; Yang et al., 2024; Zheng et al., 2024; Khaki et al., 2025). This implies that even within a single linear layer, certain parameters are far more critical for adaptation than others. However, existing LoRA-based

methods largely ignore this intrinsic structure in the parameter space, missing an opportunity for more targeted and efficient fine-tuning.

To address these issues, we propose **P**arameter-**I**mportance-Aware **Lo**w-**R**ank **A**daptation (PiLoRA), which decomposes the low-rank updates at each linear layer into distinct parameter groups based on their importance, and allocates different adaptation capacities accordingly. Our parameter partitioning operates on the output dimensions of a weight matrix, which corresponds to individual dimensions in multi-head attention layers and the internal neurons in feed-forward networks. For simplicity, we refer to this fundamental unit of partitioning as a "neuron" throughout this work. This approach facilitates a more specialized learning process with better granularity, which enhances both the effectiveness in adaptation and parameter efficiency.

Our motivation stems from a simple observation: for a pre-trained foundation model that encodes useful knowledge for downstream tasks, the importance of the learned knowledge varies across different neurons within a module. This observation also aligns with the Lottery Ticket Hypothesis (Frankle & Carbin, 2019) which states that dense networks contain small sub-networks that can be effectively trained in isolation to obtain performance comparable to the full model. Hence, it is essential to identify and focus adaptation on the most critical parameters.

To achieve this, PiLoRA first computes the gradient-based importance for different parameters at the initial phase of training. Pre-trained backbones exibit diverse feature patterns at distinct parameter positions, and parameters in even the same positions contribute differently while fine-tuning for different downstream tasks. The gradient magnitude effectively capture the varying importance of different parameters within a module. To avoid the costly computation of dense gradient for all parameters with full fine-tuning, we use the gradients on the by-default low-rank adapter to efficiently approximate this importance score. Based on the importance scores, we then identify neurons with different level of importance within each module and decompose the weight matrix into multiple components accordingly. Different adaptation capacities, i.e., LoRA ranks, are allocated to these components, with high-importance components receive higher ranks for sufficient adaptation. This gradient-informed, importance-aware allocation ensures that adaptation capacity is concentrated on where it is needed most, which results in a more efficient fine-tuning process.

We summarize our contribution as follows:

- We introduce a method to estimate parameter importance within the pre-trained weight module based on gradients from low-rank adapters, and provide visualizations of approximate gradient on the pre-trained weights.
- We propose Parameter-Importance-Aware Low-Rank Adaptation (PiLoRA), a novel PEFT method that leverages gradient-informed parameter importance for a more targeted fine-tuning process with improved memory efficiency.
- We conduct extensive experiments on various tasks to demonstrate that PiLoRA enables effective adaptation with fewer parameters.

## 2 RELATED WORK

### 2.1 PARAMETER-EFFICIENT FINE-TUNING (PEFT)

PEFT methods aim to reduce the computational cost of fine-tuning by training a small set of parameters, which can be divided into two categories: additive and selective. Additive methods introduces a small extra set of parameters in the form of LoRA (Hu et al., 2021), adapters (Houlsby et al., 2019; Chen et al., 2022) or prompts Li & Liang (2021); Jia et al. (2022) for adaptation. Selective methods identifies and trains only a sparse subset of pre-trained model's weights(Zaken et al., 2021; Sung et al., 2021; Li et al., 2022; Lu et al., 2024). For instance, Yang et al. (2024) and Zheng et al. (2024) discover critical structured components and fine-tune them exclusively with sparse activation.

### 2.2 LOW-RANK ADAPTATION

LoRA (Hu et al., 2021) employs low-rank matrices to fine-tune the pre-trained weights and inspires a line of works. One significant area of research focuses on improving the initialization of LoRA matrices to better align them with the critical subspaces of the pre-trained weights. For instance,

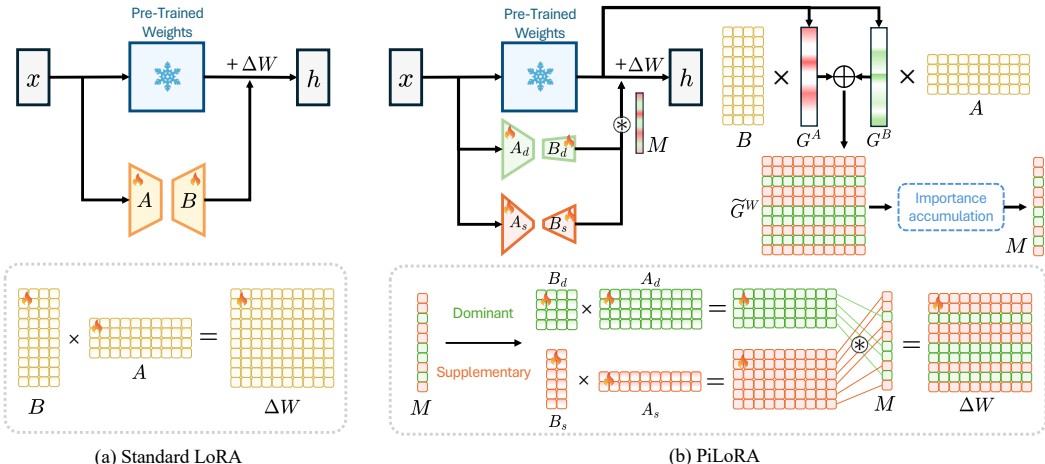

Figure 1: An overview of PiLoRA. During the importance score accumulation phase (top), the model uses LoRA gradients to approximate a gradient on the full weight, which then serves to compute neuron importance scores. These scores are then used to partition parameters into dominant and supplementary groups. In the parameter-importance-aware fine-tuning phase (bottom), these groups are trained with separate LoRAs with different rank capacity, allowing for targeted updates with parameter efficiency.

PiSSA (Meng et al., 2024) initializes the adapter by extracting the principal components of the weight matrix, while LoRA-Dash (Si et al., 2024) uses Singular Value Decomposition (SVD) to identify and focus on the most important update directions. Similarly, MiLoRA (Wang et al., 2025) proposes a strategy of merging and re-initializing LoRA weights during training to escape suboptimal local minima and enhance the optimization process. Another direction involves dynamically allocating the parameter budget during training to find the optimal rank for each module. AdaLoRA (Zhang et al., 2023) allocates the rank budget to different weight matrices based on their importance scores, while DyLoRA (Valipour et al., 2023) and ElaLoRA (Chang et al., 2025) adjust the LoRA ranks dynamically throughout training. Pruning techniques have been adapted for LoRA to further reduce its memory footprint. Methods like Zhang et al. (2024) and Zhao et al. (2024) progressively removes less salient parameters in LoRA based on gradient information or weight magnitude and reduces memory footprint during fine-tuning.

### 2.3 NETWORK SPARSITY AND PRUNING

Lottery Ticket Hypothesis proposes that for a dense network there exists small subnetworks that can be trained effectively in isolation to reach comparable performance to the full model (Frankle & Carbin, 2019). Motivated by this, the scale of LLMs has driven the development of pruning and sparse fine-tuning techniques that remove structured sparsity based on magnitude/activation (Sun et al., 2023; Khaki et al., 2025) or second-order information (Frantar & Alistarh, 2023).

## 3 METHODOLOGY

### 3.1 LOW-RANK ADAPTATION.

For a pre-trained weight matrix $W_0 \in \mathbb{R}^{m \times n}$, it is hypothesized that the updates from fine-tuning have a low intrinsic rank (Hu et al., 2021). LoRA (Hu et al., 2021) decomposes the incremental weight update $\Delta W$ into the product of two low-rank matrices:

$$W = W_0 + \Delta W = W_0 + BA,$$

where $A \in \mathbb{R}^{r \times n}$, $B \in \mathbb{R}^{m \times r}$ and $r \ll \min(m, n)$. The output of a layer is modified as:

$$h = W_0 x + \Delta W x = W_0 x + BA x.$$

$W_0$ is frozen during training and only the low-rank matrices $A$, $B$ are trainable, thus greatly reducing the computation cost.

## 3.2 Gradient-Informed Parameter Importance

Existing parameter-efficient fine-tuning (PEFT) methods, such as AdaLoRA, often assess parameter importance at a coarse level, for instance, across entire layers or modules. This approach treats all parameters within a given module as a whole and overlooks the varying importance of parameters within a weight matrix.

The gradient of a parameter fundamentally expresses the influence of a parameter, reflecting on the loss, in the fine-tuning process. In full fine-tuning, the relationship between a change in weights ($dW$) and the change in loss ($dL$) is given by the first-order approximation:

$$dL = \left\langle \frac{\partial L}{\partial W}, dW \right\rangle_F = \sum_i \sum_j \frac{\partial L}{\partial W_{ij}} dW_{ij}, \tag{1}$$

where $\langle \cdot, \cdot \rangle_F$ denotes the Frobenius inner product. For effective training, the update $dW$ must be aligned with the negative gradient $-\frac{\partial L}{\partial W}$. This implies that the magnitude of the gradient for a specific parameter, i.e., $|\frac{\partial L}{\partial W_{ij}}|$, serves as a direct measure of its importance in weight update. A large gradient magnitude indicates that the parameter undergoes greater changes to fit the downstream tasks. We can therefore define an importance score $S^* \in \mathbb{R}^{m \times n}$ for the pre-trained weight with the gradient magnitude:

$$S^* = |\frac{\partial L}{\partial W}|, \quad \text{where} \quad S_{i,j}^* = \frac{\partial L}{\partial W_{i,j}}. \tag{2}$$

**Low-rank gradients for full parameters.** While obtaining gradients w.r.t. the full parameters $W$ is straightforward in full fine-tuning, it becomes a significant challenge in PEFT with LoRA. Ideally, one would avoid the expensive computation of the gradient on $W$ via full fine-tuning. However, within the LoRA framework, the pre-trained weight $W_0$ is frozen, meaning that the gradients with respect to $W$ is not directly accessible. And only the gradients for the low-rank learnable parameters $A$ and $B$ are calculated.

To estimate importance in the full parameter space, we compute equivalent low-rank gradients w.r.t. $W$ induced by LoRA, bridging adapter updates and full-parameter updates. Note that we do not approximate the gradients produced by full fine-tuning. Although LoRA only optimizes $A$ and $B$, their gradients update $W$ via $\Delta W$, allowing transfer of gradients on $A$ and $B$ to equivalent low-rank gradients on $W$.

These equivalent low-rank gradients can serve as a computationally feasible proxy to estimate the importance of the full parameters. Since the product $BA$ approximates the full weight update $\Delta W$, the gradients on the LoRA matrices contain valuable information about the implicit gradient $\frac{\partial L}{\partial W}$ on $W$. We can therefore reconstruct an approximation of the full gradient using the adapter gradients, denoted as $G^A = \frac{\partial L}{\partial A}$ and $G^B = \frac{\partial L}{\partial B}$.

For a gradient descent step with learning rate denoted as $\eta$, and gradients $G^A$, $G^B$ are computed during backpropagation, the matrices $A$ and $B$ are updated as follows:

$$A_{t+1} = A_t - \eta G_t^A, \quad B_{t+1} = B_t - \eta G_t^B. \tag{3}$$

The change in the weight update, $\delta W$, following step t can be expressed as (see Appendix A.2 for the full proof):

$$\delta W = (B_t A_t - \eta B_t G_t^A - \eta G_t^B A_t + \eta^2 G_t^B G_t^A) - B_t A_t. \tag{4}$$

We can simplify the expression by canceling the $B_t A_t$ terms:

$$\delta W = -\eta(B_t G_t^A + G_t^B A_t) + \eta^2 G_t^B G_t^A. \tag{5}$$

Since the learning rate $\eta$ is typically a small value, the second-order term $\eta^2$ is negligible:

$$\delta W \approx -\eta(B_t G_t^A + G_t^B A_t). \tag{6}$$

This effective update, $\delta W$, is analogous to the update $-\eta \frac{\partial L}{\partial W}$ in full fine-tuning. Hence, we can define the approximate gradient on $W$ from the LoRA updates as:

$$\widetilde{G}^W \triangleq BG^A + G^B A. \tag{7}$$

Finally, we define our gradient-informed importance score $S \in \mathbb{R}^{m \times n}$ using the magnitude of this reconstructed gradient:

$$S = |\widetilde{G}^W|, \quad \text{where } S_{i,j} = |(BG^A + G^B A)_{ij}|. \tag{8}$$

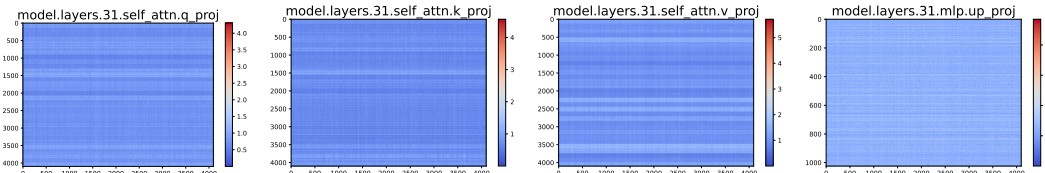

Figure 2: Visualization of approximate gradient updates on the pre-trained weight matrix. For clarity in high-dimensional MLP layers, we restrict visualization to a subset of the weights. See Figure 7 for more detailed visualizations.

---

**Algorithm 1** PiLoRA: Parameter-Importance-Aware LoRA

---

1: **Input:** Pre-trained model $\Theta_0$, Training data $\mathcal{D}_{\text{train}}$, Default rank $r$, Minimal rank $r'$, Learning rate $\eta$, Importance calculation interval $K$.
2: **Output:** Fine-tuned model $\Theta'$ with rank-differentiated LoRA modules.
                ▷ — **Stage 1: Accumulation of Importance Scores** —
3:  $M_{\text{init}} \leftarrow \text{InitializeWithStandardLoRA}(\Theta_0, r)$    ▷ Equip each module with a default LoRA
4:  For each target module, initialize hit frequency vector $H \in \mathbb{R}^m \leftarrow \mathbf{0}$.
5: **for** training step $t$ in initial phase **do**
6:   Perform forward and backward pass on $\Theta_{\text{init}}$ to compute gradients $G^A, G^B$.
7:   **if** $t \pmod K == 0$ **then**
8:    Compute $\widetilde{G}^W$ via Eq. 7
9:    Compute $S_i \leftarrow \sum_j |\widetilde{G}^W_{ij}|$ as Eq. 8) for each neuron $i = 1, \ldots, m$
10:    **for** each neuron $i = 1, \ldots, m$ **do**
11:     **if** $S_i > \bar{S}$ **then**
12:      $H_i \leftarrow H_i + 1$          ▷ Increment hit count if neuron is important
13:   Update LoRA parameters $(A, B)$ of $\Theta_{\text{init}}$ with $\eta$.
             ▷ — **Stage 2: Parameter-Importance-Aware Fine-Tuning** —
14: Create neuron mask $M$ according to $H$.
15: $\Theta' \leftarrow \text{InitializeWithMultiLoRA}(\Theta_0, m, r, r')$   ▷ Set up dominant & supplementary LoRAs
16: Let $(A_d, B_d)$ be adapters for dominant neurons (rank $r$).
17: Let $(A_s, B_s)$ be adapters for supplementary neurons (rank $r'$).
18: $\bar{M} \leftarrow 1 - M$             ▷ Create supplementary neuron mask
19: **for** each training epoch **do**
20:  **for** each batch $(x, y) \in \mathcal{D}_{\text{train}}$ **do**
21:   Forward pass with $h$ obtained from Eq. 10
22:   Compute loss $\mathcal{L}(h, y)$.
23:   Update parameters $(A_d, B_d, A_s, B_s)$ using backpropagation.
24: **return** $\Theta'$

---

### 3.3 PARAMETER-IMPORTANCE-AWARE LOW-RANK ADAPTATION (PILoRA)

To investigate the distribution of parameter importance, we visualize the approximate gradient $\widetilde{G}^W$ in the last layer of LLaMA2-7B (Touvron et al., 2023) after fine-tuning LoRA on the Commonsense 170K dataset (Hu et al., 2023), as shown in Fig. 2. A key observation is that the gradient's energy is not distributed evenly across all parameters in a weight matrix, regardless of position in the

transformer block. Instead, it concentrates in distinct horizontal bands. Since each row of the weight matrix corresponds to the parameters associated with a single output neuron, this pattern indicates that only a specific subset of neurons is important for adapting to the downstream task. The fine-tuning process predominantly modifies the weights connected to certain output dimensions while leaving others less changed. This empirical finding supports our hypothesis that different parameters within a single weight matrix are associated with highly variable importance and may require different tuning capacities. This provides a strong motivation for developing an adaptation strategy that can focus on these important parameter groups.

We propose PiLoRA, which partitions the neurons within a pre-trained weight matrix and allocates different ranks based on gradient-informed importance scores. The fine-tuning process operates in two stages: (1) an initial phase for accumulation of importance scores, and (2) a parameter-importance-aware fine-tuning phase based on the neuron partition.

**Accumulation of importance scores.** To capture the parameter importance for a given downstream task, we first equip the target weight modules with a standard LoRA and train for a brief initial phase. We periodically compute the importance score defined in Eq. 8, at fixed intervals $K$. At each interval, we identify neurons whose importance scores exceed the average at that given step and mark as important. We define a hit frequency vector $H \in \mathbb{R}^m$ where each element $H_i$ counts the number of times the $i$-th neuron was identified as important. This procedure measures not just the instantaneous importance of a neuron, but its *consistent importance* throughout the initial adaptation phase.

**Parameter-importance-aware fine-tuning.** After the initial adaptation phase, we identify important neurons based on whether the neuron is consistently engaged in the fine-tuning process as recorded by $H$. We then partition neurons within $W \in \mathbb{R}^{m \times n}$ into two groups, with important neurons indicated by a binary mask $M \in \mathbb{R}^m$:

- **Dominant group:** The important neurons (where $M_i = 1$) are assigned a high-capacity LoRA adapter, denoted by matrices $B_d \in \mathbb{R}^{|M| \times r}$ and $A_d \in \mathbb{R}^{r \times n}$ where $|M|$ is the number of important neurons, with the same rank $r$ used in the initial adaptation phase. This ensures that important parameters have sufficient capacity for effective adaptation.

- **Supplementary group:** The other neurons (where $M_i = 0$) are assigned a low-capacity LoRA adapter, denoted by matrices $B_s \in \mathbb{R}^{(m-|M|) \times r}$ and $A_s \in \mathbb{R}^{r \times n}$, with a smaller rank $r'$. By providing this low-rank adapter, we allow these neurons to still undergo small and efficient adjustments. This preserves the model's overall representational capacity with maximal parameter efficiency. Through empirical analysis, we find a small $r'$ of 1 or 2 suffices for fine-tuning these less-important neurons (Fig. 4).

To formalize the masked expansion of our compact adapters, we define the **mask mapping operator**, denoted by $\circledast$. This operator takes a binary mask $M \in \mathbb{R}^m$ and a compact matrix $B \in \mathbb{R}^{|M| \times r}$ and produces a full-dimensional matrix $\hat{B} \in \mathbb{R}^{m \times r}$. The $i$-th row of the resulting matrix is formally defined as:

$$(\hat{B})_{i,:} \triangleq (M \circledast B)_{i,:} = M_i \cdot (B)_{k(i),:} \tag{9}$$

where the index mapping function $k(i) = \sum_{j=1}^{i} M_j$ computes the corresponding row in the compact matrix $B$ by counting the number of active neurons up to row $i$. Consequently, the forward pass of a PiLoRA layer can be expressed as:

$$h = W_0 x + (M \circledast B_d) A_d x + (\bar{M} \circledast B_s) A_s x, \tag{10}$$

where $\bar{M} = 1 - M$ is the mask for the supplementary group.

Separate tuning with heterogeneous ranks ensures adaptation capacity is concentrated on the neurons that matter most, while still allowing less-important neurons to be fine-tuned efficiently with a lower capacity. Hence, PiLoRA achieves a balance between effective adaptation and parameter efficiency. The detailed algorithm is provided in Alg. 1.

Despite allocating distinct tuning capacity for neurons with different level of importance, PiLoRA can also be merged with the pre-trained weight before inference in a similar way to LoRA. With the mask mapping operator $\circledast$, the total weight update can be expressed as:

$$W' = W_0 + \Delta W$$
$$= W_0 + (M \circledast B_d) A_d + (\bar{M} \circledast B_s) A_s. \tag{11}$$

Therefore, PiLoRA introduces no architectural changes or additional latency during inference.

## 4 EXPERIMENTS

We evaluate PiLoRA across a variety of settings to validate its effectiveness, including commonsense reasoning, natural language understanding and image classification, covering 22 datasets. We compare with LoRA (Hu et al., 2021), DoRA (Liu et al., 2024), PiSSA (Meng et al., 2024), MiLoRA (Wang et al., 2025), LoRA-Dash (Si et al., 2024) and rsLoRA (Kalajdzievski, 2023). We report the percentage of parameter added and accuracy. All experiments are performed on a single NVIDIA H100 GPU. Implementation details can be found in Appendix A.4.

### 4.1 COMMONSENSE REASONING

We finetune LLaMA2-7B (Touvron et al., 2023) and LLaMA3-8B (AI@Meta, 2024) on Commonsense170K(Hu et al., 2023) and evaluate on eight tasks: BoolQ (Clark et al., 2019), PIQA (Bisk et al., 2020), SIQA (Sap et al., 2019), HellaSwag (Zellers et al., 2019), WinoGrande (Sakaguchi et al., 2021), ARC-e, ARC-c (Clark et al., 2018) and OBQA (Mihaylov et al., 2018). As shown in Tab. 1, for LLaMA2-7B, PiLoRA reduces the trainable parameter count by 24% compared to LoRA. Despite the reduction in parameters, it obtains the best performance on average, outperforming standard LoRA by 3.2% and the second-best MiLoRA by 1.6%. Similarly, for LLaMA3-8B, PiLoRA uses 21.5% less parameters than LoRA and its variants, with the best results on most tasks. This demonstrates the superiority of our importance-aware rank allocation strategy.

Table 1: Commonsense reasoning evaluation results on eight tasks. The symbol † denotes results reported in Liu et al. (2024), Wang et al. (2025) and Si et al. (2024).

| Model | PEFT | # Params(%) | BoolQ | PIQA | SIQA | HellaSwag | WinoGrande | ARC-e | ARC-c | OBQA | Avg. |
|---|---|---|---|---|---|---|---|---|---|---|---|
| ChatGPT† | – | – | 73.1 | 85.4 | 68.5 | 78.5 | 66.1 | 89.8 | 79.9 | 74.8 | 77.0 |
| LLaMA2-7B | LoRA† | 0.83 | 69.8 | 79.9 | 79.5 | 83.6 | 82.6 | 79.8 | 64.7 | 81.0 | 77.6 |
|  | PiSSA† | 0.83 | 67.6 | 78.1 | 78.4 | 76.6 | 78.0 | 75.8 | 60.2 | 75.6 | 73.8 |
|  | MiLoRA† | 0.83 | 67.6 | **83.8** | 80.1 | 88.2 | 82.0 | 82.8 | 68.8 | 80.6 | 79.2 |
|  | LoRA-Dash† | 0.83 | 71.0 | 75.7 | 79.3 | **91.1** | 78.6 | 84.2 | 69.8 | 78.8 | 78.6 |
|  | PiLoRA | 0.63 | **71.5** | 83.7 | **79.4** | 90.9 | **83.3** | **84.8** | 70.6 | **81.8** | **80.8** |
| LLaMA3-8B | LoRA† | 0.70 | 70.8 | 85.2 | 79.9 | 91.7 | 84.3 | 84.2 | 71.2 | 79.0 | 80.8 |
|  | DoRA† | 0.71 | 74.6 | 89.3 | 79.9 | 95.5 | 85.6 | 90.5 | 80.4 | 85.8 | 85.2 |
|  | PiSSA† | 0.70 | 67.1 | 81.1 | 77.2 | 83.6 | 78.9 | 77.7 | 63.2 | 74.6 | 75.4 |
|  | MiLoRA† | 0.70 | 68.8 | 86.7 | 77.2 | 92.9 | 85.6 | 86.8 | 75.5 | 81.8 | 81.9 |
|  | LoRA-Dash† | 0.70 | **75.3** | 88.5 | 80.2 | 95.7 | **86.8** | 90.7 | 80.2 | 85.6 | 85.4 |
|  | PiLoRA | **0.55** | 75.0 | **90.0** | 81.1 | **95.8** | 86.7 | **91.0** | 82.4 | 86.2 | **86.0** |

### 4.2 NATURAL LANGUAGE UNDERSTANDING

We finetune RoBERTa-large (Liu et al., 2020) on the GLUE (Wang et al., 2018) benchmark. As shown in Tab. 2, PiLoRA consistently outperforms LoRA and other baseline methods on all 7 tasks. While the baseline methods already adopt a relatively low rank for fine-tuning RoBERTa-large, PiLoRA further reduces the trainable parameter size by 18.5% with enhanced performance, showing the effectiveness of PiLoRA on natural language understanding tasks.

Table 2: Evaluation results for RoBERTa-large on 7 GLUE tasks. The symbol † denotes results reported in Fan et al. (2025).

| Method | # Params (%) | CoLA | SST-2 | MRPC | QQP | MNLI | QNLI | RTE | Average |
|---|---|---|---|---|---|---|---|---|---|
| LoRA† | 4.00 | 83.41 | 95.64 | 83.33 | 90.06 | 89.00 | 93.28 | 84.47 | 88.46 |
| DoRA† | 4.00 | 85.33 | 95.99 | 84.07 | 91.24 | 89.52 | 93.54 | 84.48 | 89.17 |
| PiSSA† | 4.00 | 69.12 | 95.98 | 82.84 | 91.24 | 88.94 | 93.59 | 73.29 | 85.00 |
| MiLoRA† | 4.00 | 84.65 | 96.10 | 86.02 | 91.33 | 89.51 | 94.12 | 84.83 | 89.51 |
| rsLoRA† | 4.00 | 83.51 | 95.98 | 86.02 | 90.75 | 88.97 | 93.84 | 84.12 | 89.03 |
| PiLoRA | 3.26 | **85.52** | **96.44** | **88.24** | **91.40** | **89.94** | **94.20** | **84.84** | **90.05** |

## 4.3 IMAGE CLASSIFICATION

We fine-tune the image encoder of CLIP ViT-B/32 (Radford et al., 2021) on 7 image classification datasets, including Cars (Krause et al., 2013), DTD (Cimpoi et al., 2014), EuroSAT (Helber et al., 2019), GTSRB (Stallkamp et al., 2012), RESISC45 (Cheng et al., 2017), SUN397 (Xiao et al., 2010) and SVHN (Netzer et al., 2011). The results in Tab. 3 shows that PiLoRA significantly reduces the gap between LoRA to full fine-tuning. With 19% fewer trainable parameters, PiLoRA achieves a notable improvement of 4.77% over LoRA and 4.37% over DoRA, with a performance close to that of full fine-tuning reported in Fan et al. (2025). This demonstrates that PiLoRA's strategy of targeted fine-tuning on dominant neurons leads to highly effective adaptation.

Table 3: Results on StanfordCars, DTD, EuroSAT, GTSRB, RESISC45, SUN397 and SVHN. The symbol † denotes results reported in Fan et al. (2025).

| Method | # Params (%) | Cars | DTD | EuroSAT | GTSRB | RESISC45 | SUN397 | SVHN | Average |
|---|---|---|---|---|---|---|---|---|---|
| **Full FT**[†] | 100 | 60.33 | 73.88 | 98.96 | 98.30 | 93.65 | 53.84 | 96.78 | 82.25 |
| **LoRA**[†] | 1.49 | 41.02 | 70.15 | 98.66 | 96.51 | 90.38 | 47.51 | 95.39 | 77.09 |
| **DoRA**[†] | 1.49 | 40.75 | 71.91 | **98.89** | 97.71 | 90.19 | 47.54 | 95.46 | 77.49 |
| **PiSSA**[†] | 1.49 | 40.41 | 69.62 | 98.48 | 95.84 | 90.58 | 47.21 | 95.84 | 76.85 |
| **MiLoRA**[†] | 1.49 | 39.77 | 70.48 | 98.19 | 97.52 | 89.92 | 45.38 | 95.49 | 76.68 |
| **PiLoRA** | **1.21** | **55.38** | **73.40** | 98.59 | **98.17** | **93.22** | **57.75** | **96.49** | **81.86** |

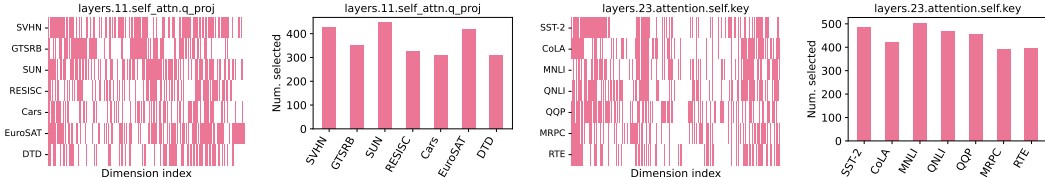

Figure 3: Visualization of dimension partition pattern in the last layer of ViT and RoBERTa-large.

## 4.4 ABLATION STUDY

**Analysis of parameter partitioning.** Fig. 3 shows the pattern of how parameters are partitioned into the dominant and supplementary groups in the last layer of ViT and RoBERTa-large across various vision and language datasets. We observe that the partition pattern vary across datasets given their intrinsic distributional differences, reflecting the distinct gradient responses they receive during training. Datasets with similar content or domain distributions tend to exhibit partially overlapping partition patterns. For instance, scene-centric datasets such as SUN and RESISC show closer patterns compared to object-centric datasets like Cars or GTSRB, while in GLUE, language inference tasks (MNLI and QNLI) also show partially shared patterns. We also find that the number of parameters being partitioned into the dominant group is relatively stable, with most tasks activating slightly less than half of the parameters. This suggests that while the specific subset of important parameters is task-dependent, the overall number of parameters actively engaged in adaptation is comparable across different tasks.

**Ablation on partition strategy.** We conduct an ablation study to validate the effectiveness of our gradient-based partition strategy for rank allocation with LLaMA2-7B in Tab. 4. We test two baselines: (1)"Random": Randomly partition neurons and assign adaptation capacity yields a poor performance, which come from insufficient adaptation on important parameters. (2)"Top-half": Partition top 50% of neurons with the highest importance scores into the dominant group improves substantially, which demonstrates the effectiveness of our gradient-informed signals for identifying important neurons. Our proposed method, which dynamically determines the partition, obtains a slightly better performance with fewer parameters. This further shows the effectiveness of our method in reducing parameters while maintaining the performance.

**Effect of varying LoRA rank for the supplementary group.** In Fig. 4, we conduct experiments to examine how the rank $r'$ assigned to the supplementary groups influences performance. While one

Table 4: Ablation on partition strategies.

| Method | # Params(%) | BoolQ | PIQA | SIQA | HellaSwag | WinoGrande | ARC-e | ARC-c | OBQA | Avg. |
|--------|-------------|-------|------|------|-----------|------------|-------|-------|------|------|
| Random | 0.66 | 62.2 | 72.7 | 50.1 | 78.1 | 47.3 | 69.1 | 56.0 | 62.0 | 62.2 |
| Top-half | 0.66 | **72.4** | **84.1** | **79.7** | 90.4 | 82.7 | 84.6 | 70.4 | 81.6 | 80.7 |
| Ours | **0.63** | 71.5 | 83.7 | 79.4 | **90.9** | **83.3** | **84.8** | **70.6** | **81.8** | **80.8** |

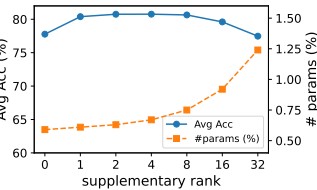
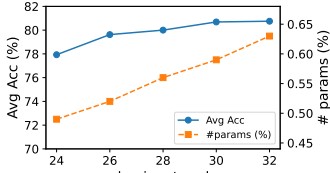
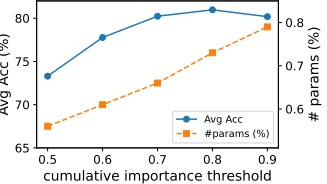

Figure 4: Effect of varying rank for supplementary group.

Figure 5: Effect of reducing rank for dominant group.

Figure 6: Effect of dominant group size.

might assume these less-important neurons could be completely frozen (i.e., $r' = 0$), this results in a notable performance drop. We hypothesize that providing at least a minimal adaptation capacity is beneficial, aligning with findings that in modern models without using ReLU non-linearities, small parameter changes can still be impactful. The results in Fig. 4 supports our hypothesis that increasing $r'$ from 0 to 1 boosts the performance as it recovers the influences by parameters in the supplementary group. The performance quickly saturates with increasing rank as the performance gain for tuning the low-importance parameter groups with more parameters is relatively marginal. Interestingly, further increasing the rank of the supplementary group to the same as the dominant one yields suboptimal performance, which suggests that over-parameterization on low-importance parameter groups may introduce noisy updates, which emphasizes the importance of targeted fine-tuning.

**Effect of varying LoRA rank for the dominant group.** Fig. 5 shows the effect of varying the LoRA rank assigned to the dominant group. While the supplementary parameters are tuned with a small rank (e.g., $r' = 2$), we observe that moderately lower ranks for the dominant group, such as 28 and 30, still achieve competitive performance close to the rank of 32 used in the initial adaptation phase. This reinforces our hypothesis that careful rank allocation based on parameter importance is more efficient than applying a high uniform rank across all parameters in a weight matrix. On the other hand, reducing the rank to much (e.g., 24) degrades performance as it limits the adaptation capacity for important parameters.

**Ablation on the size of the dominant group.** In Fig. 6, we investigate the effect of how the size of the dominant group affects performance on LLaMA2-7B. We control the group size by setting a cumulative importance threshold. A higher threshold includes more neurons in the dominant group, which in turn increases the number of trainable parameters. We observe that allowing full training capacity for more parameters generally improves performance, but the gains gradually saturate once the threshold exceeds 0.7. This indicates that a moderate subset of high-importance parameters suffices to capture most of the adaptation needs, while allocating additional capacity to the remaining parameters yields only marginal improvements. These results further highlight the efficiency of our method in concentrating adaptation capacity where it matters most.

## 5 CONCLUSION

In this work, we address a key limitation of existing LoRA-based methods that allocate the same adaptation capacity for all parameters within a weight module. We demonstrate that the uniform rank assignment is suboptimal, as it fails to leverage the varying importance of parameters within each module and the inherent structured sparsity of foundation models. We introduce PiLoRA, a novel approach that incorporates gradient-based parameter importance awareness into the PEFT fine-tuning process. By partitioning parameters into distinct groups with different adaptation capacity based on gradient-informed importance, PiLoRA strikes a balance between targeted adaptation and parameter efficiency. Experimental results on both language and vision benchmarks demonstrate the effectiveness of our method, highlighting the benefits of moving from a coarse, module-level adaptation strategy to a more fine-grained, neuron level one.

## Reproducibility Statement

We provide descriptions of the datasets used in our experiments in Appendix A.3, and report implementation details, including hyperparameter settings for each benchmark, in Appendix A.4. We will release our code upon acceptance to ensure reproducibility.

## The Use of Large Language Models (LLMs)

We declare that LLMs are only used to aid or polish writing. Specifically, ChatGPT is used to improve the clarity and grammar of writing. Research ideation, experimental design and analysis are the original contributions of the authors.

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

# A APPENDIX

## A.1 MORE VISUALIZATION OF GRADIENT IN FULL WEIGHT MATRICES

We extend Fig. 2 with detailed visualizations of various weight matrices from different layers and positions in Fig. 7. Across all layers and positions, we observe a consistent trend of gradient concentration in neuron groups (horizontal bands). We further validate this finding by visualizing the gradients for LLaMA3-8B during fine-tuning on CommonSense170K, as shown in Figure 8. These results exhibit the same concentrated patterns, demonstrating that our observation is robust across different model architectures.

## A.2 DERIVATION OF THE APPROXIMATE GRADIENT

This section provides a detailed derivation for the approximate gradient, $\widetilde{G}^W$, as defined in Equation 7. The goal is to show how the gradients of the low-rank LoRA matrices, $G^A$ and $G^B$, can be used to reconstruct an approximate gradient for the full weight matrix $W$.

Let the total weight matrix at a training step $t$ be defined by the LoRA formulation:

$$W_t = W_0 + \Delta W_t = W_0 + B_t A_t, \tag{12}$$

where $W_0$ represents the frozen pre-trained weights. The trainable LoRA matrices $A_t$ and $B_t$ are updated via gradient descent with learning rate $\eta$:

$$A_{t+1} = A_t - \eta G_t^A, \tag{13}$$

$$B_{t+1} = B_t - \eta G_t^B. \tag{14}$$

We begin by defining the change in the full weight matrix over a single training step, $\delta W$, as the difference between $W_{t+1}$ and $W_t$.

$$\begin{aligned}
\delta W &= W_{t+1} - W_t \\
&= (W_0 + \Delta W_{t+1}) - (W_0 + \Delta W_t) \\
&= B_{t+1} A_{t+1} - B_t A_t \\
&= (B_t - \eta G_t^B)(A_t - \eta G_t^A) - B_t A_t \\
&= (B_t A_t - \eta B_t G_t^A - \eta G_t^B A_t + \eta^2 G_t^B G_t^A) - B_t A_t.
\end{aligned} \tag{15}$$

Since the learning rate $\eta$ is typically a small value, the second-order term $\eta^2 G_t^B G_t^A$ is negligible in comparison to the first-order terms. We can therefore approximate $\delta W$ by dropping this term:

$$\delta W \approx -\eta(B_t G_t^A + G_t^B A_t). \tag{16}$$

The approximate change, $\delta W$, mirrors the low-rank update on the full weight $W$. In standard fine-tuning, the weight update is given by $-\eta G^W$, where $G^W = \frac{\partial \mathcal{L}}{\partial W}$. By equating the approximate expression with the standard update, we have:

$$-\eta G^W \approx -\eta(B_t G_t^A + G_t^B A_t). \tag{17}$$

This allows us to define an approximate gradient on the full parameters, $\widetilde{G}^W$, which is reconstructed from the low-rank gradients:

$$\widetilde{G}^W \triangleq B G^A + G^B A. \tag{18}$$

## A.3 DATASETS

We provide a description for the datasets we use to train and evaluate our model on commonsense reasoning, natural language understanding and image classification.

**Commonsense Reasoning:**

1. **CommonSense170K** contains 170,000 multiple-choice questions that require understanding of everyday situations, object properties, and causal relationships.

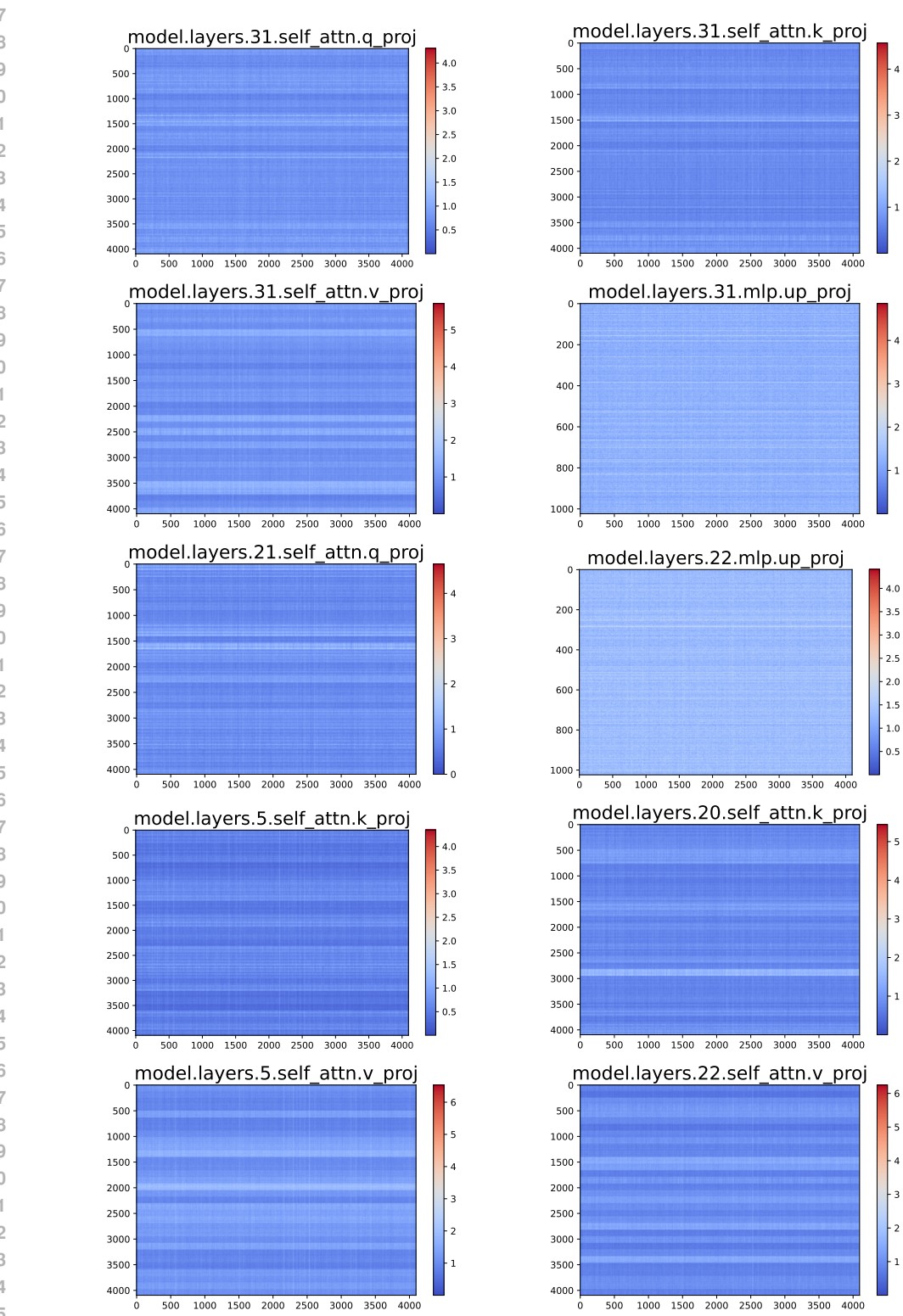

Figure 7: Visualization of approximate gradient updates on the pre-trained weight matrix for LLaMA2-7B. For MLP layers with high dimensionality, we restrict visualization to a subset of the weights for clarity.

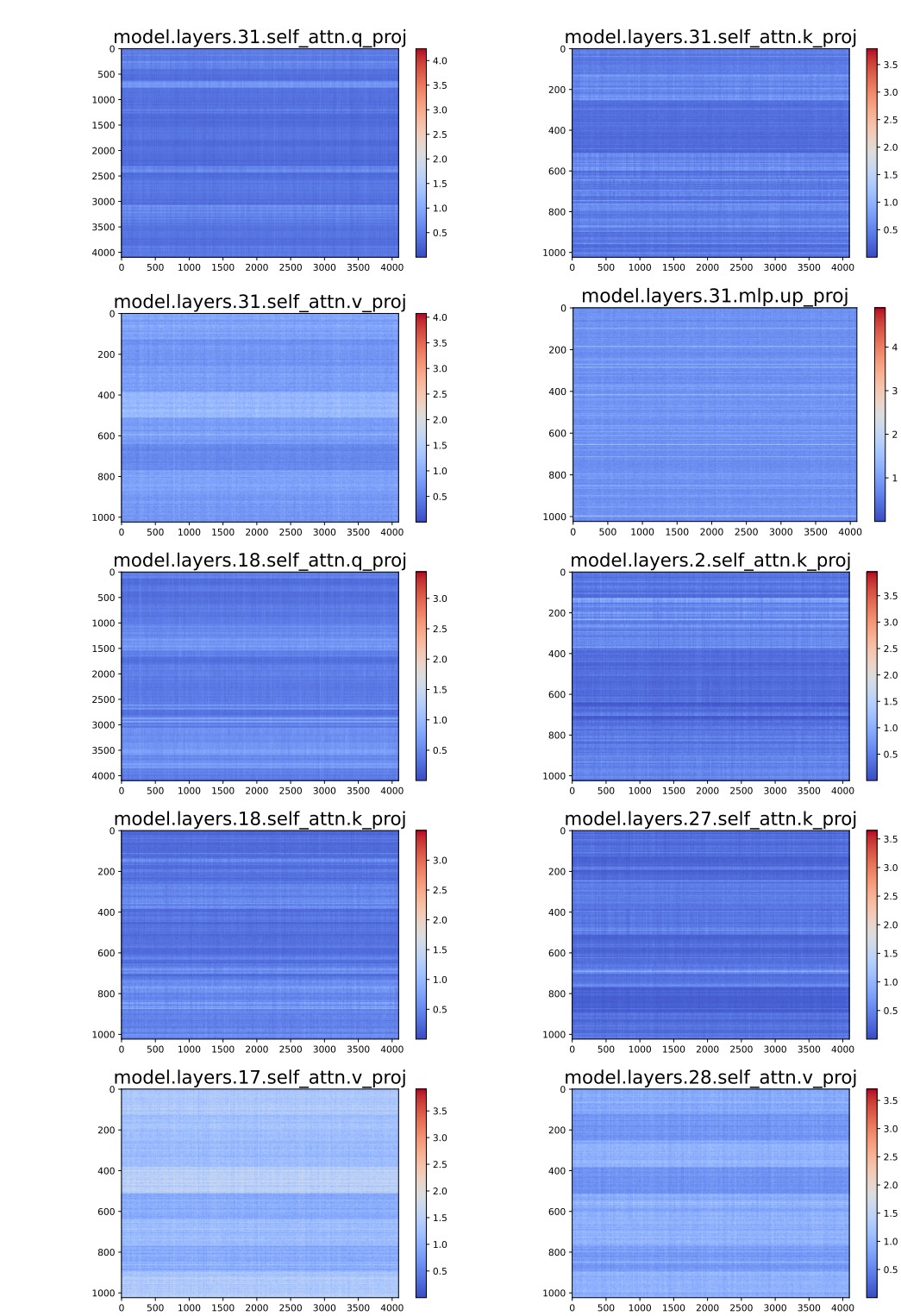

Figure 8: Visualization of approximate gradient updates on the pre-trained weight matrix for LLaMA3-8B. For MLP layers with high dimensionality, we restrict visualization to a subset of the weights for clarity.

2. **BoolQ** is a question-answering dataset containing binary questions. Each example includes a short passage from a Wikipedia article and a question that can be answered with "yes" or "no" based only on the information in the passage.

3. **PIQA** is a commonsense reasoning benchmark focused on physical interactions. It asks the model to choose the more plausible solution out of two options for a given problem.

4. **SIQA** evaluates commonsense reasoning about social situations.

5. **HellaSwag** is designed to evaluate commonsense natural language inference. The task is to complete a sentence by picking the most plausible ending from four choices.

6. **WinoGrande** focuses on pronoun resolution that requires commonsense reasoning. A model is given a sentence with a pronoun that is ambiguous and must choose the correct noun it refers to.

7. **ARC-e** is a question-answering dataset containing elementary-level science questions, which are formed as multiple-choice tasks.

8. **ARC-c** contains complex, multi-step reasoning science questions that are hard to answer with simple algorithms.

9. **OBQA** involves open-book question-answering with set of elementary science facts as the "open book".

**Natural Language Understanding:**

1. **CoLA** is a binary classification task to determine the grammatical acceptability of a sentence.

2. **SST-2** is a sentiment analysis task is to classify whether sentences contain positive or negative sentiment.

3. **MRPC** is a paraphrase detection task to determine whether the two sentences are semantically equivalent.

4. **QQP** evaluates semantic similarity with a pair of questions from the Quora website and to check for duplicates.

5. **MNLI** is a large-scale natural language inference (NLI) dataset with text from ten diverse genres.

6. **QNLI** is a NLI task with question-answering problems. The task is to determine whether a given sentence contains the answer to a given question.

7. **RTE** is a textual entailment task with a smaller dataset where the goal is to determine if one text snippet logically entails another.

**Image Classification:**

1. **Cars** (Krause et al., 2013) is a benchmark for fine-grained image classification. It contains 16,185 images of 196 classes of cars.

2. **DTD** (Cimpoi et al., 2014) is a collection of 5,640 images used for texture classification with 47 categories.

3. **EuroSAT** (Helber et al., 2019) focuses on land use and land cover classification using satellite imagery. It consists of 27,000 labeled images from the Sentinel-2 satellite, divided into 10 classes.

4. **GTSRB** (Stallkamp et al., 2012) is used for traffic sign classification. It contains over 50,000 images of 43 different traffic signs in Germany.

5. **RESISC45** (Cheng et al., 2017) is used for aerial scene classification. It contains 31,500 high-resolution images across 45 scene classes.

6. **SUN397**SUN397 (Xiao et al., 2010) is a large-scale benchmark for scene recognition, with over 100,000 images covering 397 different scene categories.

7. **SVHN** (Netzer et al., 2011) is a real-world image dataset for digit recognition. It consists of over 600,000 cropped images of house numbers obtained from Google Street View.

### A.4 IMPLEMENTATION DETAILS

All experiments are performed on a single NVIDIA H100 GPU. We report the hyperparameter setting in Tab. 5 for finetuning LLaMA2-7B and LLaMA3-8B on commonsense reasoning tasks (Liu et al., 2024). For natural language understanding and image classification tasks, We report the hyperparameter setting in Tab. 6 and Tab. 7 for finetuning RoBERTa-large (Liu et al., 2020; Fan et al., 2025) and the image encoder of CLIP ViT-B/32 (Radford et al., 2021; Fan et al., 2025). In practice, we execute the initial adaptation phase for accumulation of importance scores in 1 epoch for all settings.

Table 5: Our hyperparameter configuration on commonsense reasoning.

| Hyperparameters | LLaMA2-7B | LLaMA3-8B |
|---|---|---|
| Rank r | 32 | 32 |
| $r'$ | 2 | 2 |
| $\alpha$ of LoRA | 64 | 64 |
| Dropout | 0.05 | 0.05 |
| Optimizer | AdamW | AdamW |
| LR | 2e-4 | 1e-4 |
| LR Scheduler | Linear | Linear |
| Batch size | 16 | 16 |
| Warmup Steps | 100 | 100 |
| Epochs | 3 | 3 |
| Placement | Q, K, V, Up, Down | |

Table 6: Hyperparameters of the natural language understanding tasks.

| Hyperparameter | CoLA | SST-2 | MRPC | QQP | MNLI | QNLI | RTE |
|---|---|---|---|---|---|---|---|
| Rank | | | | 8 | | | |
| $r'$ | | | | 1 | | | |
| Alpha | | | | 16 | | | |
| Dropout | | | | 0.05 | | | |
| Optimizer | | | | AdamW | | | |
| LR | 2e-4 | 1e-4 | 1e-4 | 2e-4 | 1e-4 | 1e-4 | 2e-4 |
| LR Scheduler | | | | Cosine | | | |
| Warmup Steps | | | | 100 | | | |
| Batch Size | | | | 64 | | | |
| Epochs | 10 | 10 | 10 | 10 | 10 | 10 | 50 |

Table 7: Hyperparameters of the image classification task.

| Hyperparameter | Cars | DTD | EuroSAT | GTSRB | RESISC45 | SUN397 | SVHN |
|---|---|---|---|---|---|---|---|
| Rank | | | | 8 | | | |
| $r'$ | | | | 1 | | | |
| Alpha | | | | 16 | | | |
| Dropout | | | | 0.05 | | | |
| Optimizer | | | | AdamW | | | |
| LR | | | | 1e-4 | | | |
| LR Scheduler | | | | Cosine | | | |
| Warmup Steps | | | | 100 | | | |
| Batch Size | | | | 64 | | | |
| Epochs | 35 | 76 | 12 | 11 | 15 | 14 | 4 |

