# OpenReview forum: "PiLoRA: Gradient-Informed Parameter-Importance-Aware Low-Rank Adaptation"
_ICLR.cc/2026/Conference — ICLR 2026 Conference Withdrawn Submission_

### Official Review · Reviewer_c7Wx · 2025-10-26

**Soundness:** 2
**Presentation:** 3
**Contribution:** 1
**Rating:** 2
**Confidence:** 4

**Summary:**

To reduce model fine-tuning cost, authors introduce a new method called PiLoRA, which first use gradients to identify "more important" and "less important" neurons, and then low-rank adapters are applied to these two group of neurons with different ranks per their importance. Experiments are conducted to demonstrate the effectiveness of the proposed method.

**Strengths:**

1. This paper present a new model fine-tuning method, which first use gradients to identify "more important" and "less important" neurons, and then low-rank adapters are applied to these two group of neurons with different ranks per their importance.

2. PiLoRA consistently demonstrates stronger performance than standard LoRA and other variants while using fewer trainable parameters in commonsense reasoning, natural language understanding, and image classification tasks.

3. Ablation study and analysis is good and convincing.

**Weaknesses:**

1. Unsound preconditions derived from previous works in paper's introduction, i.e., "*It is well-established that not all neurons or parameter groups contribute equally to a model’s capabilities and structured sparsity can be used to improve the training and inference efficiency*" (Page 1, Line 50-52), where authors cite several previous works here. However, these papers do not give a decisive and agurable support for the conclusion. For instance, sparsity presented in Deja Vu is due to the relu (models are from the OPT family), and SparseLoRA merely suggests the presence of contextual sparsity (and gives the limitation about structured sparsity in their related works).

2. The method seems just like a composition of "Pruning Metric“ (see *Pruner-Zero: Evolving Symbolic Pruning Metric from scratch for Large Language Models*, ICML'24) and LoRA. The contribution is limited and seemly incremental. For me, the motivation is not good enough and authors should justify their motivation, novelity, and contribution as well.

3. The evaluation is not comprehensive enough considering the proposed method aims at overall model performance improvement. Authors should conduct summarization and math reasoning experiments with LLaMA-3-8B to evaluate the text generation. Additionally, authors should also evaluate the image generation performance to better demonstrate that the proposed model works in most common tasks. My concern here is: models are only required to generate the next token in the experiments in this paper, i.e., commonsense reasoning, natural language understanding, and image classification, thus it needs more experiments to address this concern.

**Questions:**

See Weakness. I'll consider raising the score if authors can address the weakness.

---

### Official Review · Reviewer_JDpw · 2025-10-27

**Soundness:** 2
**Presentation:** 2
**Contribution:** 1
**Rating:** 2
**Confidence:** 5

**Summary:**

This paper introduces PiLoRA, a parameter-efficient fine-tuning method that allocates different adaptation ranks to neurons based on their gradient-informed importance. Unlike LoRA, which applies the same rank to all modules, PiLoRA estimates neuron importance using low-rank gradient approximations and assigns higher ranks to more important neurons. The approach improves efficiency and performance across language and vision tasks, showing consistent gains over existing LoRA variants while reducing trainable parameters.

**Strengths:**

- Easy to read and well-organized.
- Proposes a clear and intuitive gradient-informed importance estimation for rank allocation.
- Includes comprehensive ablation studies and visualizations.

**Weaknesses:**

1. This paper proposes a rank allocation method based on gradient-informed importance, but it does not clearly compare its approach with prior studies that use similar strategies, such as ElaLoRA [1]. In particular, although ElaLoRA is included in the references, the authors do not explicitly acknowledge that their method is also gradient-based, nor do they clearly describe the conceptual or experimental distinctions from those prior works.
2. The proposed method appears to assign rank $r$ to important neurons and a predefined smaller rank $r′$ to less important ones. This structure seems a specific form of ElaLoRA, where ranks are dynamically allocated based on gradient-based importance. Therefore, the authors should more clearly explain how their method differs from ElaLoRA and what its unique contributions are.
3. Comparisons with some of the latest methods are missing. For example, ElaLoRA [1],  AdaLoRA [2], DyLoRA [3], and GoRA [4] (I'm not sure that [4] does essential in ICLR 26 review phase, but a preprint is first uploaded to arXiv in February 2025) are not included. This omission reduces the reliability of the performance evaluation.
4. Equations (3)–(7) appear to describe concepts already proposed in LoRA-GA [5] and its subsequent studies. Therefore, it is difficult to regard them as the paper’s original methodology. These equations would be more appropriately placed in a Preliminaries or Background section rather than in the Method section.
5. Table 1 appears to include some values cited from the AdaLoRA paper. If that is the case, it would be more convincing to also present results under the same experimental conditions (e.g., $r=2$, $r=8$). Furthermore, it is somewhat questionable that AdaLoRA [2], a major adaptive rank allocation method, is excluded from the baseline comparison.
6. Were Tables 1–3 produced from a single run? In general, especially for NLU tasks, multiple runs are important, but standard deviation values are not reported.
7. Missing runable source code.

---

> [1] Chang, Huandong, et al. "Elalora: Elastic & learnable low-rank adaptation for efficient model fine-tuning." *arXiv preprint arXiv:2504.00254* (2025).
>
> [2] Zhang, Qingru, et al. "Adalora: Adaptive budget allocation for parameter-efficient fine-tuning." ICLR 2023
>
> [3] Valipour, Mojtaba, et al. "DyLoRA: Parameter-Efficient Tuning of Pre-trained Models using Dynamic Search-Free Low-Rank Adaptation." *Proceedings of the 17th Conference of the European Chapter of the Association for Computational Linguistics*. 2023.
>
> [4] He, Haonan, et al. "Gora: Gradient-driven adaptive low rank adaptation." *arXiv preprint arXiv:2502.12171* (2025).
>
> [5] Wang, Shaowen, Linxi Yu, and Jian Li. "Lora-ga: Low-rank adaptation with gradient approximation." *Advances in Neural Information Processing Systems* 37 (2024): 54905-54931.

**Questions:**

please refer to the Weaknesses

---

### Official Review · Reviewer_tDCE · 2025-10-28

**Soundness:** 2
**Presentation:** 2
**Contribution:** 2
**Rating:** 2
**Confidence:** 5

**Summary:**

This paper introduces a new parameter efficient fine tuning method that seeks to allocate different adaptation capacity to the fine tuners based on parameter importance. In order to approximate the importance of parameters during training the authors use low-rank gradients that partition the neurons into distinct groups. Some neurons are then shown to be more important to others and thus receive a higher adaptation strategy for targeted training, while the less important neurons are tuned with lower adaptation capacity. This leads to an efficient fine tuning program. The authors show that their methodology performs effectively on both language reasoning and image classification tasks.

**Strengths:**

**Originality:** The method of the paper is interesting and original as applying a strategy that puts importance on different neurons for fine tuning has not been done before as far as I am aware. It leads to a nice way to efficiently remove the importance on those neurons that are not having much of an affect on the gradient of the network and redirects that importance to other neurons.

**Clarity:** The paper is generally written well with mostly clear explanations on the core idea and insight. I would have liked to see a theoretical explanation of why their method yields better performance. I understand that this approach can lead to a more efficient fine tuning algorithm but why does this methodology yield better downstream performance?

**Weaknesses:**

**Novelty:** While the idea of the paper is interesting I don't think it is very novel. Importance strategies have been used previously in deep learning and while they have managed to come up with such a strategy for gradients in the fine tuners of a PEFT model the authors don't seem to give any insight into why this works the way it does on PEFT models. Furthermore, the authors do not include any limitations of their work.

**Significance:** The paper would be much more significant if the authors would have provided some theoretical insight into why their methodology yields better performance. Their methodology suggests that importance should be split across different weight parameters making up the fine tuning adaptors which is very reasonable. It is then reasonable to expect that this would yield better convergence than a general LoRA strategy as now the gradient trajectory should be more optimal for convergence. Yet the authors don't make any comment or remark to this effect let alone give any insight on the convergence. After all, their whole paper is rooted in providing importance to gradients and yet there is no mention of how this affects the actual gradient optimization they then employ.

**Questions:**

1. Could you provide some theoretical insight into why PiLoRA is able to yield better downstream performance than usual baselines? By providing importance to different weight parameter gradients I would expect better training. Why does that translate to better downstream performance?

2. By putting different importance on parameter gradients you should see better convergence as the gradients are only used for training. Is it the case that PiLoRA also results in better convergence of the adaptors in the sense that adaptors converge faster? Is it possible to provide a theoretical argument that shows that this is in fact the case? This would really help strengthen the paper.

3. The idea of putting importance on different parameter gradients seems like it should also be applicable in the case of pre-training a model. Is this the case? For a large model, couldn't you apply a similar idea to PiLoRA and group the fully dense weights into two groups, one where you give higher importance to the parameter gradients of that group and the other where you give lower importance?

4. What are the limitations of PiLoRA?

---

### Official Review · Reviewer_KvFW · 2025-11-01

**Soundness:** 3
**Presentation:** 3
**Contribution:** 3
**Rating:** 4
**Confidence:** 4

**Summary:**

PiLoRA is a novel method for Parameter-Efficient Fine-Tuning (PEFT), addressing some of the limitations in existing LoRA-based methods. While LoRA has become widely adopted for fine-tuning pre-trained models efficiently by injecting low-rank matrices and keeping the pre-trained weights frozen, it treats the weights in a module uniformly, which can be suboptimal. PiLoRA seeks to improve this by introducing a finer-grained approach that takes into account the varying importance of parameters within a module.

PiLoRA uses a gradient-based method to compute the importance of parameters during the fine-tuning process, partitioning them into "important" and "supplementary" neurons. The "important" neurons receive more adaptation capacity, while less critical neurons are updated with lower capacity, optimizing computational resources while maintaining performance.

**Strengths:**

PiLoRA significantly reduces the number of parameters needing fine-tuning by focusing adaptation on the most important neurons, resulting in lower computational cost.

The gradient-informed importance scores enable a more targeted fine-tuning process, allowing PiLoRA to outperform vanilla LoRA with fewer parameters.

Unlike other PEFT methods that operate at the module level, PiLoRA performs neuron-level adaptation, which leads to better utilization of the adaptation budget.

Empirical results across language and image classification tasks show that PiLoRA outperforms existing methods like LoRA, DoRA, and others while using fewer parameters.

**Weaknesses:**

The gradient-based importance estimation adds some complexity to the training process compared to standard LoRA.

While the initial phase to calculate importance scores is efficient, it still adds extra steps in the training pipeline, which could be a disadvantage in time-sensitive applications.

**Questions:**

Language coverage includes GLUE/Commonsense but lacks long-context, code models, etc.
Add evaluations for: math reasoning, code, and vision-language tasks.

Importance is determined only during the warm-up phase, not updated dynamically.
Refer to DyLoRA for progressive re-ranking or online mask updates.

---

### Note · Authors · 2025-11-18

I have read and agree with the venue's withdrawal policy on behalf of myself and my co-authors.